# One-Pot Self-Assembly of Core-Shell Nanoparticles within Fibers by Coaxial Electrospinning for Intestine-Targeted Delivery of Curcumin

**DOI:** 10.3390/foods12081623

**Published:** 2023-04-12

**Authors:** Lijuan Hou, Laiming Zhang, Chengxiao Yu, Jianle Chen, Xingqian Ye, Fuming Zhang, Robert J. Linhardt, Shiguo Chen, Haibo Pan

**Affiliations:** 1National-Local Joint Engineering Laboratory of Intelligent Food Technology and Equipment, Zhejiang Key Laboratory for Agro-Food Processing, Integrated Research Base of Southern Fruit and Vegetable Preservation Technology, Zhejiang International Scientific and Technological Cooperation Base of Health Food Manufacturing and Quality Control, College of Biosystems Engineering and Food Science, Zhejiang University, Hangzhou 310058, Chinachenshiguo210@163.com (S.C.); 2Innovation Center of Yangtze River Delta, Zhejiang University, Jiaxing 314102, China; 3Center for Biotechnology and Interdisciplinary Studies, Rensselaer Polytechnic Institute, Troy, NY 12180, USA

**Keywords:** coaxial electrospinning, self-assembled, intestine-targeted, curcumin

## Abstract

Nanotechniques for curcumin (Cur) encapsulation provided a potential capability to avoid limitations and improve biological activities in food and pharmaceutics. Different from multi-step encapsulation systems, in this study, zein–curcumin (Z–Cur) core-shell nanoparticles could be self-assembled within Eudragit S100 (ES100) fibers through one-pot coaxial electrospinning with Cur at an encapsulation efficiency (EE) of 96% for ES100–zein–Cur (ES100–Z–Cur) and EE of 67% for self-assembled Z–Cur. The resulting structure realized the double protection of Cur by ES100 and zein, which provided both pH responsiveness and sustained release performances. The self-assembled Z–Cur nanoparticles released from fibermats were spherical (diameter 328 nm) and had a relatively uniform distribution (polydispersity index 0.62). The spherical structures of Z–Cur nanoparticles and Z–Cur nanoparticles loaded in ES100 fibermats could be observed by transmission electron microscopy (TEM). Fourier transform infrared spectra (FTIR) and X-ray diffractometer (XRD) revealed that hydrophobic interactions occurred between the encapsulated Cur and zein, while Cur was amorphous (rather than in crystalline form). Loading in the fibermat could significantly enhance the photothermal stability of Cur. This novel one-pot system much more easily and efficiently combined nanoparticles and fibers together, offering inherent advantages such as step economy, operational simplicity, and synthetic efficiency. These core-shell biopolymer fibermats which incorporate Cur can be applied in pharmaceutical products toward the goals of sustainable and controllable intestine-targeted drug delivery.

## 1. Introduction

Curcumin (Cur) is a natural active substance extracted from the tubers of herbs such as Curcuma longa [1], Rhizoma zedoariae [2] and Radix curcumae [3], and it has broad application prospects in the fields of foods [4,5] and medicines [6,7]. It has a strong coloration ability and various physiological effects including anti-oxidation, anti-inflammation [8], anti-tumor [9], bacteriostasis, and promoting wound healing [10]. However, several factors, including the insolubility [11], low stability at light and thermal changes [12], and low in vivo bioavailability [13] of free Cur in water largely affect its application to be even more. The construction of a nanoscale transport system is an effective solution to solve this problem [14,15]. Clinical studies have found that systems with nanocarriers can significantly enhance Cur’s bioavailability and stability and therefore make it more applicable [16].

Zein is a natural protein macromolecule extracted from corn endosperm. Its two key features are its good biocompatibility and self-assembly characteristics; therefore, zein is an ideal and safe raw material in food production [17,18] and medicines [19,20] and has attracted widespread attention. Nanoparticles (NPs) along with zein are used as carrier materials to enhance the stability and bioavailability of insoluble drugs [21]. However, zein is not an ideal nanocarrier material due to its low stability, burst release, and encapsulation efficiency [22]. Therefore, researchers have studied to identify these inadequacies as well as identify more desirable qualities of this macromolecule (e.g., pH responsiveness) [23].

Eudragit S100 (ES100) is a pH-sensitive synthetic polymer of which the upper limit of the pH threshold is 7. Since 2010, it has had increasingly wide application in drug industries [24,25]. ES100 is promising in delivering drugs because it is not only insoluble in aqueous media but also nontoxic, nonirritant, biocompatible and permeable, which make drug release controllable. Given that ES100 is characterized by gastro resistance, gastrointestinal targeting, and an effective sustained release of drugs, it becomes an ideal option for targeting colorectal diseases [26,27].

In a variety of nanocarriers, the electrospinning nanofiber delivery system has a simple preparation process and low cost; some common drugs and even some protein molecules can be added into the electrospinning solution. Electrospinning has been proposed as a feasible way for drug delivery and bioactive compounds encapsulation [28,29,30]. The release rate and cumulative release amounts of drugs in vivo can be controlled by changing the process parameters of electrospinning and the material composition in the spinning fluid [31]. The drug distribution in the fibers prepared by electrospinning technology is relatively uniform and can, to a large extent, avoid the deactivation of drug components [32].

The studies have examined a range of delivery systems which are used for the encapsulation of Cur, including a biopolymer-based system, lipid-based system, and emulsion-based system [33]. Most of them involve multiple steps for processing such as nanoprecipitation first and then incorporation into hydrogel [34] or a combined nanoemulsion technique with a high-speed homogenizer and ultrasonic probe [35]. Under these circumstances, synthesizing multicomponent copolymers with tailored structures, ensuring quality control, achieving reproducible manufacturing, and obtaining validated characterization all pose challenges for large-scale production. As a result, there remains an unmet demand for developing simple, cost-effective, and robust approaches with excellent scalability and consistency for manufacturing polymeric nanoplatforms that offer versatile functions and a wide range of applications [36]. The hypothesis of this study is that the zein–curcumin (Z–Cur) core-shell NPs could be self-assembled within ES100 fibers through a one-step coaxial electrospinning method without any multi-step process required, and Cur had an encapsulation efficiency (EE) of 96%. The resulting structure would afford the double protection of Cur by ES100 and zein which provided both pH responsiveness and sustained release performances. Encapsulated Cur and zein form self-assembled zein–curcumin (Z–Cur) NPs through hydrophobic interactions. Z–Cur NPs were spherical (diameter 328 nm), and narrowly distributed (polydispersity index 0.62). Loading in a fibermat could significantly enhance the photothermal stability of Cur. In addition, Z–Cur NPs which were encapsulated in ES100 fibermat had been proven effective in delaying the speed of releasing Cur in simulated gastrointestinal fluids. The aim of this work is that it can provide an idea for the design of vectors; this novel one-pot system much more easily and efficiently combines nanoparticles and fibers together. Access to such a system through a one-pot method offers inherent advantages such as step economy, operational simplicity, and synthetic efficiency [37]. In addition, these core-shell biopolymer fibermats could be incorporating Cur into pharmaceutical products for intestine-targeted sustained and controlled drug delivery.

## 2. Material and Methods

### 2.1. Materials

The Eudragit S100 was kindly provided by Evonik (Darmstadt, DE). Zein (98.0%) was purchased from J&K Scientific Ltd. (Beijing, China), ethanol absolute (≥99.5, ACS reagent grade) and curcumin (≥95.0% purity) were purchased from Sigma Chemical Co. (St. Louis, MO, USA).

### 2.2. Electrospinning Procedure

Regarding the fabrication of ES100–zein–Cur and ES100–Cur fibermats, to prepare ES100–zein–Cur (ES100–Z–Cur) mats, 20 mg of zein and 20 mg of Cur were dissolved in 10 mL of 85% (*v/v*) ethanol as the core solution, and 14% (*w/v*) ES100 was dissolved in ethanol absolute as the shell solution. The ES100–Cur fibermat was selected as control and the preparation procedure was the same as that for ES100–Z–Cur in addition to replacing the core solution with 20 mg of Cur dissolved in 5 mL of 85% (*v/v*) ethanol. The coaxial fibers were produced through utilizing a coaxial-electrospinning setup (MECC, Ogori, Fukuoka, Japan). The radius of the inner needle was 0.575 mm and that of the outer needle was 0.8 mm. The spacing between the aluminum collector electrode and spinneret tip was 9 cm with an applied voltage of 10 kV. The flow rate of core solutions was 0.4 mL/h, and that of the shell solution was 0.8 mL/h. All samples were electrospun below 50% relative humidity.

### 2.3. Characterizations of NPs and Fibermats Formation

#### 2.3.1. SEM and TEM Observation

A field-emission scanning electron microscope (FE–SEM) (FEI–Versa, Hillsboro) was employed to monitor the morphological evolution of fibermats. For a better SEM monitoring, the samples were covered by a film of palladium and gold through sputter-coating. Before cross-sectioning, the fibermats should be broken in liquid nitrogen [10]. Transmission electron microscopy (TEM) was observed by a JOEL JEM 1400 PLUS (JEOL Ltd., Japan). Briefly, the Z–Cur nanoparticles were dispersed in phosphate buffer solution (PBS) and then dropped on a copper grid. For the fibermat observation, the copper grid was fixed on the collector and fibermat around 2 s during electrospinning. The copper grids of nanoparticles or fibermats were observed with the accelerated voltage of 80 kV [9].

#### 2.3.2. Particle Size and Zeta-Potential Measurements

Drawn on methods outlined in the existing literature [38] yet modified in service to our own research purposes, we used ZS Zetasizer Nano (Malvern Instrument Ltd., Malvern, UK) to measure parameters including the z-average particle diameter, polydispersity index (PDI) and zeta potential of self-assembled Z–Cur NPs. The NPs were dispersed in PBS (0.01 M, pH 7.2) at a 1:25 (*w*/*v*). The polydispersity index (PDI) and z-average particle were measured by dynamic light scattering (DLS) on a Zetasizer Nano-ZS90 (Malvern Instruments, Worcestershire, UK). The level of scattered light was detected at a 90° angle. After equilibrating liquid samples for 60 s at 25 °C, they were processed by dynamic light back scattering [39]. The data were collected based on ten succeeding readings. Finally, the zeta potential for dispersions is measured in a distinct electric field in terms of the velocity and direction of particle movement.

#### 2.3.3. Encapsulation and Loading Efficiency

Drawing from methods in an earlier study [40], a UV-vis spectrophotometer was employed to measure the quantity of Cur which had been loaded and encapsulated in ES100–Z–Cur fibermats and self-assembled Zein–Cur (Z–Cur) NPs. The standard solutions of Cur had created a calibration curve which indicates good linearity (r^2^ = 0.9998). The Z–Cur NPs were released from ES100–Z–Cur via phosphate-buffered saline (PBS) (0.01 M, pH 7.2) before the stage of centrifugation. Z–Cur, ES100–Cur and ES100–Z–Cur fibermats were dissolved by absolute ethanol. We used a UV–1800 spectrophotometer (Shimadzu Corporation, Kyoto, Japan) to measure the specific contents of Cur. Absorbance was determined at a 426 nm wavelength. The loading efficiency (LE) and encapsulation efficiency (EE) were calculated by the following equations:EE (%) = Encapsulated curcumin/Total curcumin input × 100
LE (%) = Encapsulated curcumin)/Weight of fibermats × 100

### 2.4. FTIR Measurement

Approximately 1 mg of each sample was ground and pressed into pellets with 200 mg potassium bromide. A Nicolet iN10 (Thermo Fisher Scientific, Waltham, MA, USA) instrument was utilized for Fourier transform infrared (FTIR) measurement with the scanning frequency set to 4000–400 cm^−1^ [39].

### 2.5. XRD Measurement

Based on graphite–monochromated Cu Kα radiation results from a Bruker D8-discover X-ray diffractometer (XRD), the crystal transformation of Cur can be observed [27].

### 2.6. DSC Measurement

A differential scanning calorimeter (DSC) 8500 (Perkin-Elmer) was used to measure the thermal properties of fibermats. Pre-weighed fibermats were encapsulated in aluminum pans and were heated at 10 °C/min from 20 to 300 °C under nitrogen condition [39].

### 2.7. TGA Measurement

Thermogravimetric analysis (TGA) was conducted on a TGA Q50 apparatus, using computer-controlled TA instruments, under a nitrogen atmosphere when the samples were heated from room temperature to 800 °C [41].

### 2.8. Photostability and Thermostability Measurements

Drawn on the method described by [39,42] but modified in service to this research, we investigated the impacts of carriers on Cur against UV photolysis in order to measure the stability of free Cur and encapsulated Cur in fibermats. In total, pure Cur, Z–Cur, ES100–Cur and ES100–Z–Cur were placed in flasks and sat in a light box (Q-Sun, Q-Lab Corporation, Westlake, OH, USA) for two hours; then, they were dissolved in absolute ethanol with a final concentration of 10 μg/mL. A spectrophotometry analysis was conducted to measure the amount of remained Cur. The evaluation of thermostability followed a similar procedure in addition to cultivating samples in a water bath for two hours with a starting temperature of 80 ℃ before it was cooled down to room temperature (25 °C).

### 2.9. In Vitro Cur Release in NPs and Fibermats

Under simulated gastrointestinal (SGI) conditions [43], we evaluated the properties of releasing Cur in Z–Cur, ES100–Cur, ES100–Z–Cur and free Cur. Since the solubility of Cur in water is remarkably low, the water containing 0.05% Tween 80 is a relatively good medium for testing the release profile [44]. More specifically, simulated gastric fluid (SGF) and simulated intestinal fluid (SIF) were mixed with absolute ethanol in equal amounts. Then, a given quantity of sample (30 mg) was kept in the dialysis bag of which the molecular cut-off was set at 3 kDa. The sample was kept in a flask at 37 °C containing 30 mL of SGF release medium and was shaken gently for two hours. Then, the dialysis bag containing the samples was moved to the flask and kept for three hours under the SIF conditions. The UV-vis spectrophotometer was used to measure the Cur content of the release medium. To keep the volume unchanged, an additional fresh release medium with the same volume was added into the flask.

### 2.10. Statistical Analysis

Each operation was carried out in triplicate. Data are displayed in the format of mean values ± standard deviations (SD). Statistical analysis was conducted on Origin 9.0 through running a one-way ANOVA with Tukey’s test. A *p*-value ≤ 0.05 is statistically significant.

## 3. Results and Discussion

### 3.1. Formation of the Core-Shell Nanoparticles within Fibers

The preparation diagram is demonstrated in Figure 1. The obtained ES100–Z–Cur fibermat was yellow colored (Figure 2A). Scanning electron microscopy (SEM) images of self-assembled Z–Cur NPs loaded in ES100 fibermats are presented in Figure 2B,C. SEM images show that cylindrical, flat and extremely thin ES100–Z–Cur nanofibers were produced (Figure 2B). The SEM cross section and transmission electron microscopy (TEM) clearly showed the core-shell structure of the fiber, and the NPs were distributed in the fiber (Figure 2C,D). The diameter value of these nanofibers was around 600 nm (Figure 2D). The self-assembled Z–Cur NPs in the ES100 fibermats were released by PBS with pH = 7.2, and these NPs were spherical and had a core-shell structure (Figure 2E,F). The appropriate proportion of Z–Cur NPs that can self-assemble within the ES100 fiber and evaluation of NPs stability were also investigated by size distribution, polydispersity index (PDI) and zeta potential analysis (see Appendix A); with the increase in zein–Cur ratio from 20 mg–20 mg (20–20) to 40 mg–20 mg (40–20) in electrospun core solution, the average diameter of self-assembled Z–Cur NPs changed from 328 to 900 nm (Appendix A), and the self-assembled 40–20 NPs were difficult to encapsulate and only arranged on the surface of the ES100 fiber (Appendix A). Meanwhile, the PDI value of 20–20 was 0.62 and gradually increased from 20–20 to 40–20 (Appendix A), and the potential zeta stability decreased from 20–20 to 40–20 (Appendix A), which means 20–20 had a relatively uniform particle size distribution and better stability than 40–20. This phenomenon is attributed to the mechanism of zein self-assembly. The enhancement of solvent polarity will lead to the decrease in solubility of zein and the change of its molecular conformation, leading to molecular aggregation. Zein self-assembly is induced by slowly evaporating ethanol in the solvent to form NPs. In this process, the concentration of zein is positively correlated with the particle size [45,46]. Therefore, we only selected the 20–20 Z–Cur ratio for the study; the Cur had an EE of 96% in ES100–Z–Cur fibermats, EE of 67% in self-assembled Z–Cur, and LE of 1.4% for Cur in ES100–Z–Cur fibermats.

### 3.2. FTIR Analysis

The FTIR spectra of pure Cur, zein, ES100, self-assembled Z–Cur, ES100–Cur, and ES100–Z–Cur are presented in Figure 3. In regard to free Cur, no characteristic peaks were observed, which indicates that Cur was in the form of keto-enol tautomeric form [47]. The spectrum of ES100 reveals sharply characteristic peaks of carbonyl groups (C=O) at 1727 cm^−1^ and characteristic bands of hydroxyl groups (C-H stretch vibration) 2957 cm^−1^. Additional spectra, 1152 cm^−1^ and 3087 cm^−1^, also indicated C-O and O-H stretch vibration, and the strong absorption peak at 3400 cm^−1^ corresponds to O-H stretching vibration [48]. The spectral range of zein show characteristic peaks of amide I and amide II at 1651 and 1523 cm^−1^, which were caused by C=O stretching vibrations, N-H bending, and C-N stretching vibrations [49]. The characteristic peaks of Cur (1602, 1509, 1429, 1279, 1151, 964 and 857 cm^−1^) by and large disappeared in the spectra of the fibermats, which were caused by the non-covalent interactions (hydrophobic effects) between Cur and zein. Our research results reaffirm most of the existing studies in which the characteristic peaks of polyphenols ceased to be visible when they were encapsulated in composite carrier [50,51]. When the Z–Cur NPs were encapsulated in ES100 fibermats, the FTIR of the formulation demonstrated the same pattern, in which there were no remarkable changes in spectra as well as interactions between NPs and ES100. It revealed that Cur or self-assembled Z–Cur NPs were encased in the ES100 fibermats.

### 3.3. XRD Analysis

The stability and solubility of functional components are determined by their crystalline state. X-ray diffraction analysis (XRD) is widely used to test the crystallinity of encapsulated compounds or biopolymer matrix. The XRD analysis (Figure 4) of pure Cur in our research is consistent with previous studies, which demonstrates a set of unique characteristic peaks at 2θ of 8.84, 14.42, 17.23, 19.33, 21.05, 23.31, 24.47, 25.53, and 28.96 [42]. These indicators prove their crystalline structures [52]. In addition, zein showed two broad peaks at 2θ of 9 and 19, which is closely related to the amorphous proteins [41,42]. The XRD pattern of encapsulated Cur in the NPs, however, was distinct from that of pure Cur, and there were no characteristic crystal peaks of Cur. The disappearing characteristic crystal peaks confirm that Cur was in an amorphous form when it was encapsulated in the protein matrix. The combination of Cur and protein not only consequently affects the ability of crystalizing Cur in the nanostructure, it also reduced the ability of crystallization by becoming the amorphous complex. Due to the higher oral bioavailability observed in the amorphous state than that in the crystalline one, the mixture of Cur and protein has the potential for improving the oral administration of Cur within the delivery system [53,54]. Due to the non-crystalline form of the blank ES100, there were no distinctive diffraction peaks observed when the Cur or Z–Cur NPs was encapsulated in ES100 fibermats, which is also in line with the FTIR results.

### 3.4. DSC Analysis

The changing physical state of bioactive compounds was reflected on a differential scanning calorimeter (DSC) (Figure 5). The DSC curve of pure Cur revealed a sharp endothermic peak at the temperature of 177 °C, which was caused by the melting of Cur crystals. This peak proved that Cur was in a crystalline state [55]. It is remarkable that the characteristic endothermal peak of Cur was not detected in Z–Cur, ES100–Cur and ES100–Z–Cur, which suggested that Cur was widely distributed within the NPs and fibermats in amorphous form. The XRD analysis shown in Figure 4 reaffirmed these findings, and it indicated that Cur forms an amorphous complex with ES100 and zein.

### 3.5. Thermal Performance Analysis

The thermogravimetric analysis (TGA) showed the disintegration patterns and thermal stabilities of Cur, zein and ES100 (Figure 6 and Table 1). In the first phase, surface-absorbed water evaporation led to reducing 5% weight mass of zein, 4% of Z–Cur, 6% of ES100, 11% of ES100–Cur and 10% of ES100–Z–Cur. In contrast to ES100–Cur, the additional zein in ES100–Z–Cur was closely related to reducing the ability of absorbing water. In phase two, molecular chain fracture and thermal decomposition led to reducing 58% of the weight mass of Cur from 246 to 420 °C, 70% of zein and Z–Cur from 240 to 400 °C, 87% of ES100 from 366 to 450 °C, 80% of ES100–Cur from 375 to 450 °C, 78% of ES100–Z–Cur from 383 to 450 °C. The working temperature of the maximum rate of weight reduction and terminating decomposition temperature of ES100–Z–Cur fibermats were higher than those of ES100–Cur fibermats, which indicated that zein had improved the thermal stability of ES100–Z–Cur fibermats. This is consistent with the research results of [41,56].

### 3.6. Photostability and Thermostability Analysis

Cur is sensitive to a range of environmental factors such as oxygen, light and heat, which limit its wider application. In this study, we examined the stability of encapsulated Cur under conditions of ultraviolet light and thermal treatment [57]. In contrast to the absorbance values of free Cur, Cur encapsulated in ES100 and zein NPs was much higher, which means it was stable when exposed to UV irradiation. After being exposed to UV irradiation for 120 min, over 70% of Cur remained in the ES100–Z–Cur, 45% remained in the ES100–Cur and 3.8% remained in the free Cur (Figure 7A). The double bonds in zein molecules and aromatic side groups are capable of absorbing UV light, which results in high stability when it was exposed to the light [58]. In addition, under 80 ℃ treatment for 120 min, more than 70% of Cur remained in the ES100–Z–Cur, 52% remained in the ES100-Cur and 20% remained in the free Cur (Figure 7B). Sun et al. [39] embedded Cur with zein and shellac colloidal particles, and its retention rate after heat treatment (60 °C, 30 min) was 94%, which is 3.1 times higher than that of free Cur (30%). Liang et al. [59] embedded Cur with zein/chitosan quaternary ammonium salt NPs, and its thermal stability increased 3.5 times. The additional zein strengthened the thermal stability of ES100–Z–Cur fibermats, which is consistent with TGA results. The presence of zein and ES100 both contributed to protect Cur from degrading when it is exposed to UV irradiation and thermal treatment. Our research results were consistent with previous findings reported by [57,60].

### 3.7. In Vitro Controlled Release of Cur in Different Carriers

The release properties of Cur are shown in Figure 8. We used free Cur as the control to examine the in vitro sustained release properties of drug-loaded nanocarriers. Only 17% free cur was detected in the digestive fluid during the first 120 min of SGF digestion. After 180 min SIF digestion, a total of 26% of Cur was detected in the digestive solution, indicating that the solubility of pure Cur powder was very low even in an aqueous solution containing Tween 80. The release rate of Cur in self-assembled Z–Cur NPs was the lowest among Z–Cur, ES100–Cur, ES100–Z–Cur after 300 min digestion. This might be explained by the strong hydrophobic interaction between Cur and zein [50]. For self-assembled Z–Cur NPs, around 46.71% of the Cur was released after SGF digestion and exhibited a burst release in the first 60 min. After 180 min SIF digestion, around 61.48% of the Cur was released. Because of the pH responsiveness of the ES100 polymer, ES100–Cur and ES100–Z–Cur, there was almost no release of Cur during the first 120 min of SGF digestion, while Cur was released rapidly when exposed to SIF digestion. After 180 min of SIF digestion, 80.28% and 76.45% of Cur were detected in ES100–Cur and ES100–Z–Cur, respectively. Compared with ES100–Cur, the release of Cur in ES100–Z–Cur was slower, showing better sustained release characteristics. The sustained-release characteristics of ES100–Z–Cur nanocarriers can be summarized into the following two reasons: (1) the strong hydrophobic interaction between Cur and protein prevents drug diffusion from the inside of NPs, thus prolongating the time of drug release [56], and (2) the self-assembled Z–Cur NPs are encapsulated in the ES100 fibermats, forming a second protective layer to further delay the release of Cur [24].

## 4. Conclusions

The Z–Cur core-shell NPs could be self-assembled within ES100 fibers through a one-pot coaxial electrospinning method with Cur at EE 96% for ES100–Z–Cur and EE 67% for self-assembled Z–Cur NPs. The resulting structure achieved both pH responsiveness and sustained release performances by ES100 and zein. The self-assembled Z–Cur NPs released from fibermats were spherical (diameter 328 nm) and had a relative uniform distribution (polydispersity index 0.62). The spherical structures of Z–Cur NPs and Z–Cur NPs loaded in ES100 fibermats were observed by TEM. The encapsulated Cur was in an amorphous form, which was detected by DSC and XRD diffraction. FTIR spectra revealed that hydrophobic interactions occurred between the encapsulated Cur and zein. The photothermal stability of Cur was significantly improved after being loaded in a fibermat. Furthermore, the release of Cur in simulated gastrointestinal fluids was notably delayed when Z–Cur was encapsulated in the ES100 fibermat. This study can provide an idea for the design of vectors, as this novel one-pot system much more easily and efficiently combined nanoparticles and fibers together, offering inherent advantages such as step economy, operational simplicity, and synthetic efficiency. These core-shell biopolymer fibermats which incorporate Cur can be applied in pharmaceutical products toward the goals of sustainable and controllable intestine-targeted drug delivery.

## Figures and Tables

**Figure 1 foods-12-01623-f001:**
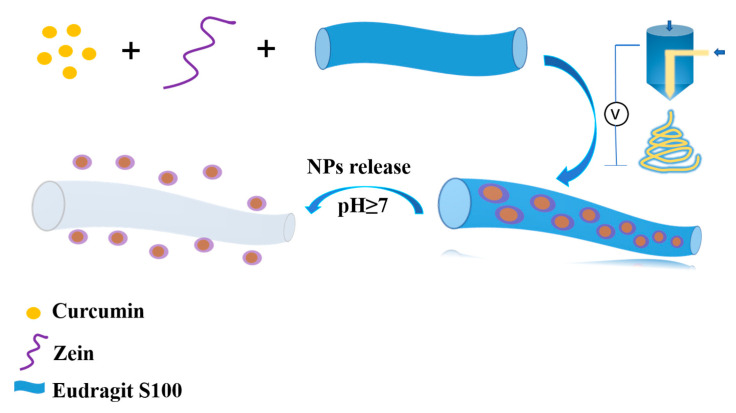
The schematic diagram of Z–Cur NPs assembly and encapsulation in fibermats by co-axial electrospinning and release of Cur.

**Figure 2 foods-12-01623-f002:**
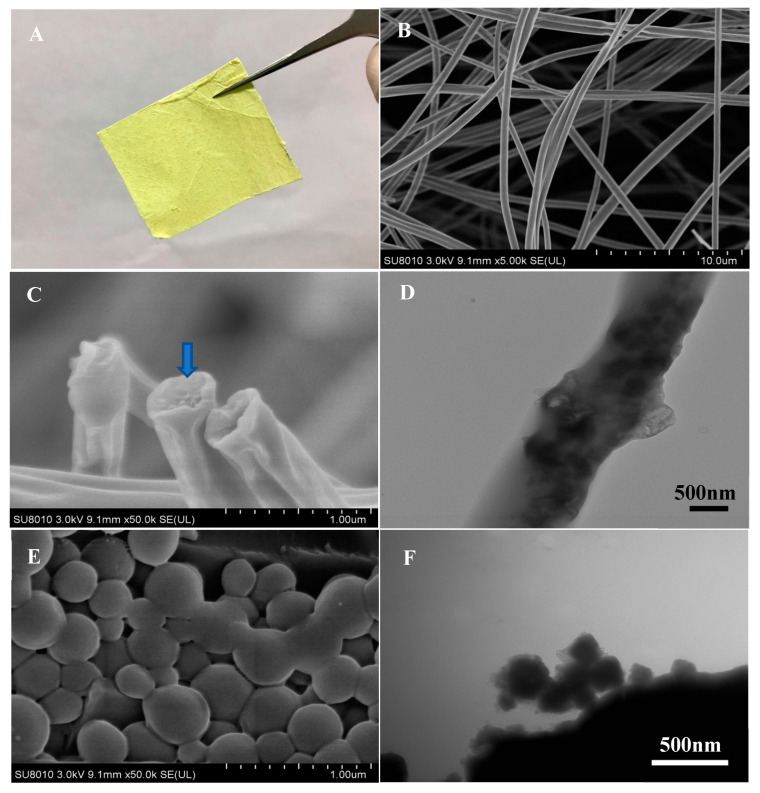
(**A**) Photograph and (**B**,**C**) SEM images and (**D**) TEM of ES100–Z–Cur. (**E**) SEM and (**F**) TEM of self-assembled Z–Cur NPs released from ES100–Z–Cur.

**Figure 3 foods-12-01623-f003:**
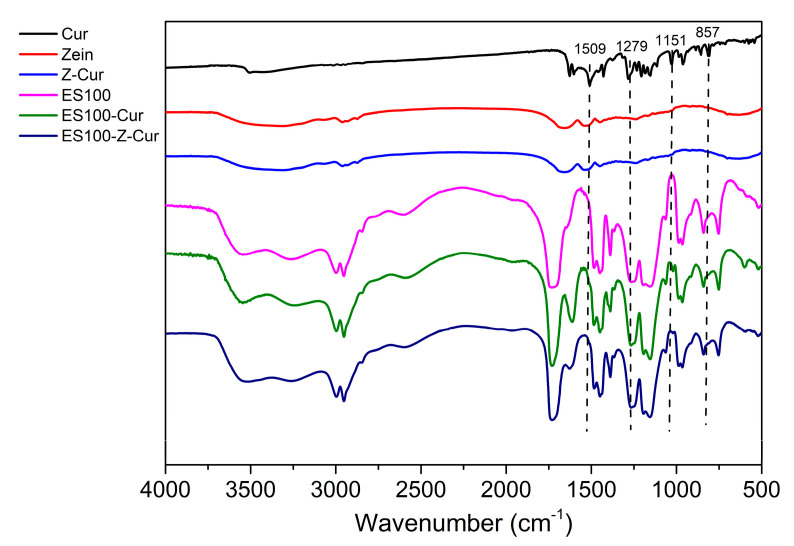
Fourier transform infrared spectroscopy of pure Cur, zein, released self-assembled Z–Cur, ES100, ES100–Cur, and ES100–Z–Cur.

**Figure 4 foods-12-01623-f004:**
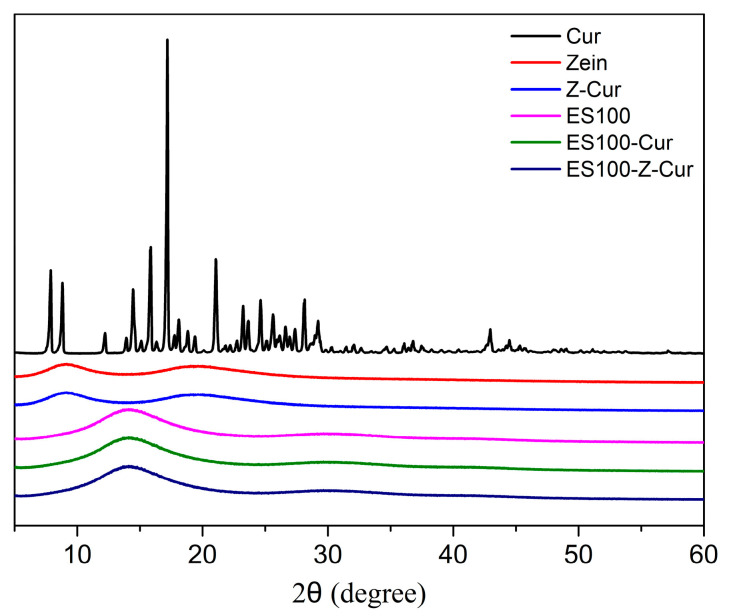
XRD patterns of pure Cur, zein, released self-assembled Z–Cur, ES100, ES100–Cur, and ES100–Z–Cur.

**Figure 5 foods-12-01623-f005:**
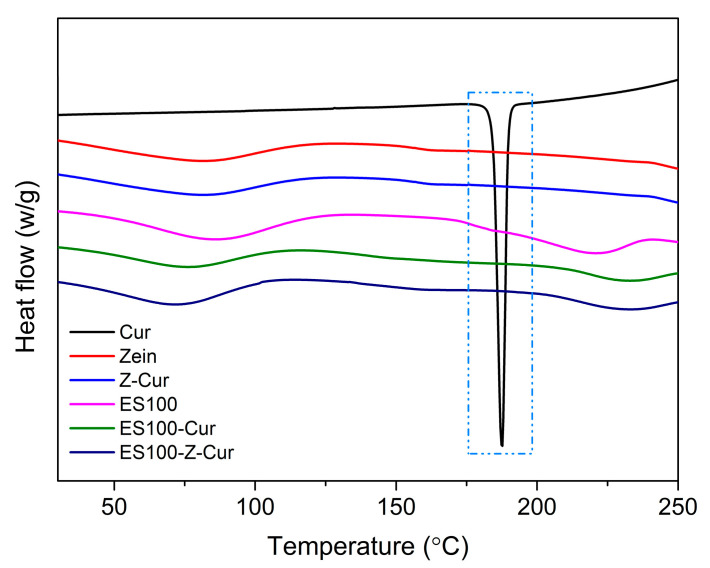
DSC thermograms of pure Cur, zein, released self-assembled Z–Cur, ES100, ES100–Cur, and ES100–Z–Cur.

**Figure 6 foods-12-01623-f006:**
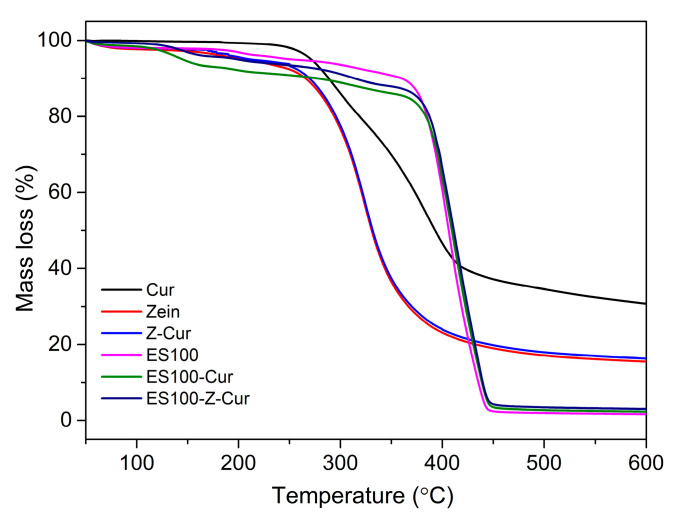
TGA curves of pure Cur, zein, released self-assembled Z–Cur, ES100, ES100–Cur, and ES100–Z–Cur.

**Figure 7 foods-12-01623-f007:**
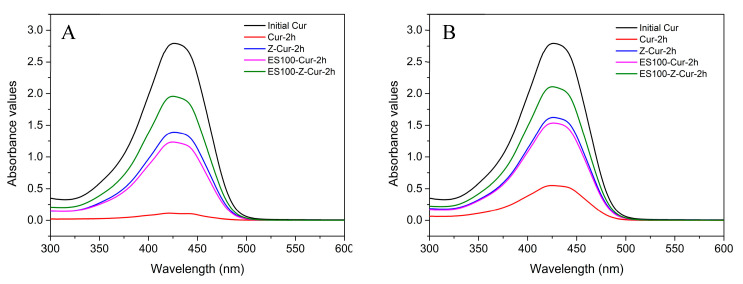
Absorption spectra of Cur, released self-assembled Z–Cur, ES100–Cur, ES100–Z–Cur prepared in ethanol after (**A**) UV treated 2 h, (**B**) 80 °C treated 2 h.

**Figure 8 foods-12-01623-f008:**
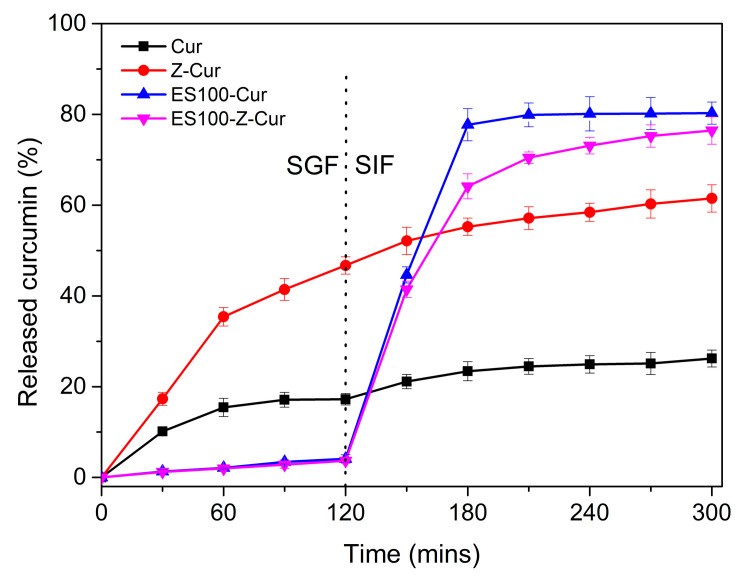
In vitro release of Cur from pure Cur, released self-assembled Z–Cur, ES100–Cur, and ES100–Z–Cur. Error bars represent the standard deviation.

**Table 1 foods-12-01623-t001:** Mass residues at different temperature range obtained from TGA curves.

Samples/Temperature (°C)	0–100	100–200	200–300	300–400	400–500	500–600
Cur	99.85	99.27	86.07	46.66	34.55	30.66
Zein	97.65	95.15	76.67	23.08	17.09	15.47
Z–Cur	97.97	95.59	77.61	23.94	17.91	16.29
ES100	98.03	96.87	93.57	60.65	1.95	1.64
ES100–Cur	98.37	92.13	88.92	63.67	2.70	2.29
ES100–Z–Cur	99.22	94.94	91.13	66.49	3.43	3.00

## Data Availability

Data is contained within the article or Appendix A.

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
