# Peer review of "One-Pot Self-Assembly of Core-Shell Nanoparticles within Fibers by Coaxial Electrospinning for Intestine-Targeted Delivery of Curcumin"

_foods, 2023, doi:10.3390/foods12081623_

Round 1

Reviewer 1 Report

You can find revision as Word file. Manuscript is fine. I do not find it fits aims and scopes of Journal Foods. Therefore, I suggest journals: International Journal of Molecular Sciences, Pharmaceuticals, Pharmaceutics.

Author Response

Dear reviewer,
Thank you for the comments concerning our manuscript entitled “One-pot self-assembly of core-shell nanoparticles within fibers by coaxial electrospinning for intestine-targeted delivery of curcumin”. Those comments are all valuable and very helpful for revising and improving our paper, as well as the important guiding significance to our research. We have studied comments carefully and have made correction that we hope will be satisfactory. Revised portions are marked in yellow in manuscript. Thanks so much!

Reviewer 2 Report

The physical properties of the designed curcumin delivery, ES100-Z-Cur, were described in this study. The potential advantage and application of ES100-Z-Cur was demonstrated in Figure 8. In general, the manuscript can be published with minor revision.

1. Please check the accuracy of the overall content, for example, line 257 should be Figure 4, not 3. Some of the descriptions may need to be revised, and too-long sentences may confuse. Please check typing errors. 

2. All the experiments compared the properties of Cur, Zein, Z-cur, ES 100, ES100-Cur, and ES100-Z-Cur. However, missing data from Z-Cur in the thermal performance analysis (Figure 6). Please check if the data of Zein was not from Z-Cur. 

3. Please indicate the scale bar in Figure 2F. Give more detail about the Cur release study and TEM observation.  

Author Response

(The authors gave the same response as above.)

Reviewer 3 Report

One-pot self assembly of core-shell nanoparticles within fibers by co-axial electro-spinning for intestine targeted delivery of curcumin

 In the manuscript titled “One-pot self assembly of core-shell nanoparticles within fibers by co-axial electro-spinning for intestine targeted delivery of curcumin” the authors have demonstrated about the use of Zein Curcumin shell core nanoparticles self assembled with Eudragit S100 fibers. Characterization studies were conducted on the Z - cur nanoparticles released from fibermats. These core shell biopolymer fibermats which incorporate Curcumin can be suggested for use in pharmaceutical products as suggested. SEM images are clear.

Major Points

 1. The authors have stated that the proposed method is a single step process. How far does this process improve the delivery system?

 2. In which way the proposed single step conversion process is advantageous than the multi-step conversion method?

 3. In Section 2.6 Encapsulation and loading efficiency the equations are given. But in Results and Discussion they are not included. Similarly particle size and zeta potential shown in separate subheadings are not found in Results and discussion section.

 4. In Materials and Methods section, the content is divided into 11 subheadings, but in Results and Discussion photo and thermal stability are concatenated and there are seven subheadings. Kindly let them to be uniform.

Minor Points

1. In the abstract, 8 line, NP may be expanded as nanoparticles as it is not included only in Page 2, Line 48.

2. In 11th line of abstract, closed parentheses “)” is missing.

3. In Page 2, Line 47, materials shall be used in singular.

4. In Page 2, Line 49 “stability of” shall be “stability” - of shall be omitted.

5. Kindly complete last sentence of section 2.3.

6. Table 1 shall be formatted for clarity.

 Based on the submission it can been seen that the authors have done work for curcumin delivery for controllable intestine targeted drug delivery. The paper may be accepted in the Journal Foods following slight revision. Figures are nice and references are quite new.

Author Response

(The authors gave the same response as above.)
